# Assessing Ethnic Minority Representation in Fibromyalgia Clinical Trials: A Systematic Review of Recruitment Demographics

**DOI:** 10.3390/ijerph20247185

**Published:** 2023-12-15

**Authors:** Peter Henley, Tanimola Martins, Reza Zamani

**Affiliations:** Medical School, College of Medicine and Health, University of Exeter, Exeter EX1 2LU, UKtom207@exeter.ac.uk (T.M.)

**Keywords:** fibromyalgia, trials, non-White, ethnic minorities, underrepresentation, pharmaceutical, retention, longitudinal

## Abstract

The under-representation of non-White participants in Western countries in clinical research has received increased attention, due to recognized physiological differences between ethnic groups, which may affect the efficacy and optimal dosage of some treatments. This review assessed ethnic diversity in pharmaceutical trials for fibromyalgia, a poorly understood chronic pain disorder. We also investigated longitudinal change to non-White participant proportions in trials and non-White participants’ likelihood to discontinue with fibromyalgia research between trial stages (retention). First, we identified relevant trials conducted in the United States and Canada between 2000 and 2022, by searching PubMed, Web of Science, Scopus, and the Cochrane Library databases. In trials conducted both across the United States and Canada, and exclusively within the United States, approximately 90% of participants were White. A longitudinal analysis also found no change in the proportion of non-White participants in trials conducted across the United States and Canada between 2000 and 2022. Finally, we found no significant differences in trial retention between White and non-White participants. This review highlights the low numbers of ethnic minorities in fibromyalgia trials conducted in the United States and Canada, with no change to these proportions over the past 22 years. Furthermore, non-White participants were not more likely to discontinue with the fibromyalgia research once they were recruited.

## 1. Introduction

The issue of the under-representation of non-White participants in clinical research has recently received increased attention, due to the recognition that differences in the efficacy of some medical treatments can be attributed to physiological differences between ethnic/racial groups [1,2]. Genetic differences between ethnic groups have been shown to affect some drugs’ metabolism, thereby influencing efficacy, tolerability, and optimal dosage [3,4,5,6]. Furthermore, the prevalence, pathogenesis, and prognosis of chronic long-term conditions, such as Alzheimer’s, cancer, and several cardiometabolic diseases, also vary considerably by ethnicity [7,8,9,10,11]. Therefore, testing the efficacy of new treatments and preventative strategies for these conditions across ethnic groups is not only critical to efficacy, but may impact adherence and outcomes [1,2,4,5,6]. 

It is widely recognized that the representation of White participants in clinical trials is higher than all other ethnic groups, primarily due to the majority of trials being conducted in Western countries [12]. Notably, changes to clinical research infrastructure and funding have increased clinical trial outputs in Asian countries over the past 20 years, with China having the highest trial output behind the United States [12,13,14]. However, globally, the representation of non-White and non-Asian ethnic groups in trials remains low, although estimates of these proportions range from 8% to 30% depending on the type of clinical research [2,15,16]. This lack of ethnic diversity has gained recent attention, with a number of government, private, and research funding organizations setting up initiatives to increase the proportion of non-White participants in clinical research [1,17,18,19,20]. For example, the US Federal Drug and Food Administration (FDA) has highlighted the lack of ethnic diversity in clinical research, and published guidelines on the need for higher numbers of non-White participants to adequately trial new treatments [1,20]. 

Fibromyalgia is an incurable chronic pain disorder, which is also associated with fatigue, sleep disturbances, cognitive problems, and mood changes [21]. Pain in fibromyalgia develops in the absence of any obvious cause, thus it is generally attributed to altered sensory processing [21,22]. Treatment focuses mainly on pain management, although this is complex, with variable outcomes [21]. Subsequently, over the past 20 years, there have been many clinical trials investigating the repurposing of existing pharmaceutical treatments for managing fibromyalgia pain and accompanying symptoms [21,22,23]. Fibromyalgia prevalence is unknown; however, it has been estimated to be around 2% [24]. It is approximately 20 times more common in females [24,25,26], and differences between ethnic groups are contentious due to potential under-diagnosis and health inequalities [27,28,29]. 

To our knowledge, the ethnicity demographics of participants in fibromyalgia research have not been investigated. This review addresses this question in the context of pharmaceutical trials conducted in the United States and Canada. We focused on the United States since, compared with other areas of the world, this region has both a more ethnically diverse population, as well as a relatively higher fibromyalgia research output. However, we also included studies from Canada and Puerto Rico for most of our objectives, to be inclusive of the largest randomized control trials, which were primarily conducted in the United States, but had some sites in Canada and Puerto Rico. To this end, we first identified clinical trials investigating pharmaceutical treatments for fibromyalgia that were conducted in these countries. We then assessed the non-White participant proportions in these trials (Objective 1), and the proportions of White, Black, and other ethnic group participants, across the trials which were exclusively conducted in the United States (Objective 2). Next, we investigated whether the proportion of non-White participants in the trials conducted across the United States and Canada have changed over the past 22 years (Objective 3), and finally whether non-White participants were more likely to withdraw from fibromyalgia research between trial stages, in other words, trial retention (Objective 4). 

## 2. Methods

### 2.1. Literature Searches

The methodology of this systematic review follows the PRISMA-DTA guidelines for systematic review and meta-analysis.

Searches for relevant articles were performed on 30 June 2022 in the PubMed, Web of Science, Cochrane Library, and Scopus databases. Keywords included fibromyalgia, clinical trial, randomized control trial, and RCT. Filters available within each database were used to eliminate the articles not meeting the inclusion criteria. For search details, see Appendix A.

### 2.2. Inclusion Criteria

Eligible studies investigated pharmacological agents for the treatment of any fibromyalgia symptoms. All articles were published in English between 1 January 2000 and 30 June 2022, and recruited participants either exclusively within or across the United States, Canada, and Puerto Rico. All studies recruited new participants (i.e., they were not continuation studies or post hoc analyses of previous studies), except for two selected studies for Objective 4. The full criteria used to specify the types of articles included in this review are provided in Appendix A.

### 2.3. Screening 

Article screening was performed using Rayyan (2022 version) [30]. The Rayyan deduplication function was used to remove exact article duplicates and flag potential duplicates. Two reviewers (PH and RZ) then screened the remaining articles, excluding those not meeting the inclusion criteria and the remaining duplicates. Articles were first screened by title, or title and abstract, to remove obvious ineligible results (e.g., reviews). Where necessary, the main text was then read to screen the remaining articles. Where access to papers was unavailable, manuscripts were requested from the corresponding authors, following which these articles were excluded if the full text was not provided.

### 2.4. Data Extraction 

Data on participants’ ethnicity and gender was extracted from all the included articles, where this information was available. In trials where this data was reported at multiple trial stages, this information was extracted for the stage that had the most participants (except for Objective 4, see ‘Analysis’ Section 2.5). 

### 2.5. Analysis

Data processing and analysis were performed in R (version 4.2.2). For all statistical tests, including those checking statistical test assumptions, a *p* value < 0.05 was deemed statistically significant. 

Selected eligible articles were included for each review objective, see below. For Objectives 1–3, the ethnicity data relevant to each of these objectives were not reported in some of the eligible studies; thus, the number of eligible trials reporting these data is reported in the results, prior to assessing the ethnic diversity in the studies where this information was available. In trials where the relevant ethnicity metadata were not reported for some of the participants, these trials were included in our pooled analysis, with the missing data points reported as ‘data not available’. Trials that presented ethnicity demographic metadata imprecisely (e.g., participants were 79–89% White) were excluded from the pooled analysis, and reported as not reporting the relevant ethnicity demographics precisely.

Objective 1—Here, we assessed the proportion of White vs non-White participants in all trials recruiting new participants. Analysis of ethnic subgroup was not possible due to the limited reporting of granular ethnic groupings in the metadata of eligible trials. 

Objective 2—This included all eligible studies conducted exclusively in the United States. Due to a slightly higher proportion of these trials reporting the numbers of Black participants, the proportion of White, Black, and other unspecified ethnic groups was assessed in these trials.

Objective 3—To assess longitudinal changes to the proportion of non-White participants in multi-centre trials recruiting new participants, Pearson’s product-moment correlation was used to test for an association between the proportion of non-White participants in the trials and year of study. To check Pearson’s correlation assumptions, the linearity of the association was tested by plotting and visually inspecting the residuals; the year of study was then transformed using the quadratic term and tested for association with the proportion of non-White participants. Normal distribution of both variables and the residuals was tested using the Shapiro Wilk test, following which only multi-centre trials were included, due to the proportion of non-White participants in single-centre studies being positively skewed and zero inflated. Outliers with a Cooks distance > 1 were to be removed, but none were identified. 

Objective 4—To assess potential differences between White and non-White participants remaining in clinical research between trial stages/retention, by identifying initial and continuation trial phases. To this end, we identified articles reporting at least the proportion of White and non-White participants at more than one trial stage. Next, we extracted this data for the earliest and latest stages within these individual studies, to use as the initial and continuation phases respectively. To identify additional continuation stages, we also identified studies that recruited all their participants from previous trials. Since some trials had more than one continuation study, we identified the continuation study that was the most recent. Participant data for all of the initial and continuation phases were then pooled separately to compare retention rates between White and non-White participants between trial stages. A Chi-squared test was then used to statistically test the differences between observed and expected White and non-White participant retention.

## 3. Results

### 3.1. Identification of Fibromyalgia Research

Our searches identified 3572 articles after duplicate removal. We excluded 3506 articles not meeting the eligibility criteria, and eight that could not be accessed, leaving 58 fibromyalgia trials [31,32,33,34,35,36,37,38,39,40,41,42,43,44,45,46,47,48,49,50,51,52,53,54,55,56,57,58,59,60,61,62,63,64,65,66,67,68,69,70,71,72,73,74,75,76,77,78,79,80,81,82,83,84,85,86,87,88] for analysis. For screening details and exclusion reasons, see the PRISMA flowchart (Figure 1). The characteristics of included trials are summarized in (Table 1).

#### 3.1.1. Study Characteristics 

Fifty-six trials [31,32,33,34,35,36,37,38,39,40,41,42,43,44,45,46,47,48,49,50,51,52,53,54,55,56,57,58,59,60,61,62,63,64,65,66,67,68,69,70,71,72,73,74,75,76,77,78,79,80,81,82,83,84,85,86] recruited new participants (Objective 1), of which 46 [31,32,33,34,35,36,37,38,39,40,41,42,43,44,45,46,47,48,49,50,51,52,53,54,55,56,57,58,59,60,61,62,63,64,65,66,67,68,69,70,71,72,73,74,75,76] were conducted exclusively in the United States (Objective 2). Five of the remaining trials were single-centre studies conducted in Canada [82,83,84,85,86], and five were multi-centre trials conducted primarily in the United States, with some sites in Canada or Puerto Rico [77,78,79,80,81]. To assess longitudinal changes to the proportion of non-White participants in clinical trials, the 27 multi-centre trials [52,53,54,55,56,57,58,59,60,61,62,63,64,65,66,67,68,69,70,71,72,76,77,78,79,80,81] that reported the relevant data were used (Objective 3). To compare White and non-White participant retention in trials (Objective 4), six studies [68,69,78,81,87,88] with relevant data were used. Two of these studies [68,69] reported the proportion of White and non-White participants in more than one phase within the same study, providing two initial phases and two continuation phases across those two studies. Additionally, two studies [87,88] were identified that were continuations of the other two initial studies [78,81] (i.e., recruited all their participants from those initial studies). Pooled together, this gave 3399 participants from the four initial trial stages, and 1674 (49.2% of the original) participants from the four continuation phases.

#### 3.1.2. Ethnic Diversity of Fibromyalgia Trials in the United States and Canada (Objective 1)

Of the 56 trials recruiting new participants [31,32,33,34,35,36,37,38,39,40,41,42,43,44,45,46,47,48,49,50,51,52,53,54,55,56,57,58,59,60,61,62,63,64,65,66,67,68,69,70,71,72,73,74,75,76,77,78,79,80,81,82,83,84,85,86], 12 reported no ethnicity demographic information. Another one of the studies that reported ethnicity demographics was additionally excluded from the final analysis, as the reporting was imprecise (i.e., White participants between 79% and 89%). Overall, 43 of the total 56 trials (76.8%) clearly reported at least the proportion of White and non-White participants. These 43 studies comprised a total of 14,214 participants, of whom 90.1% were White, 9.9% were non-White, and data were unavailable for only three participants (<0.1%) (Figure 2). In individual studies, the proportion of White participants ranged between 77.4% and 100%. 

#### 3.1.3. Ethnic Diversity of Fibromyalgia Trials Conducted Exclusively in the United States (Objective 2)

Of the 46 trials that recruited participants exclusively in the United States [31,32,33,34,35,36,37,38,39,40,41,42,43,44,45,46,47,48,49,50,51,52,53,54,55,56,57,58,59,60,61,62,63,64,65,66,67,68,69,70,71,72,73,74,75,76], 27 (58.7%) reported the numbers of White and Black participants in their metadata. Of the total 8311 participants across these 27 studies, 90.6% were White, 4.9% Black, 4.5% were from other ethnic groups, and data were unavailable for just one participant (<0.1%), (Figure 3).

#### 3.1.4. Longitudinal Assessment of Non-White Participant Representation in Trials (Objective 3)

Of the 29 multi-centre trials included for analysis [52,53,54,55,56,57,58,59,60,61,62,63,64,65,66,67,68,69,70,71,72,73,75,76,77,78,79,80,81], 27 reported the numbers of non-White participants in their metadata [52,53,54,55,56,57,58,59,60,61,62,63,64,65,66,67,68,69,70,71,72,76,77,78,79,80,81]. Across these 27 trials, 10% of the total 13,665 participants were non-White, with these proportions ranging between 4% and 22.6% in individual trials. No significant change to these proportions was identified between the start of 2000 and end of June 2022, (Pearson’s, *r* = −0.13, [95% CI: −0.49, 0.26], *p* = 0.52) (Figure 4). 

#### 3.1.5. White and Non-White Participant Retention in Trials (Objective 4)

The retention rates between the four identified initial trial stages and the four continuation stages [68,69,78,81,87,88] were 49.0% and 51.6% for White and non-White participants, respectively. There was no significant difference between the observed vs expected retention numbers between the two groups (Chi-squared test, *p* = 0.53 [Retention observed vs.|retention expected: White 1513|1520.3, non-White 161|153.7]).

## 4. Discussion

This review identified a low proportion of non-White participants in fibromyalgia research, with a White participant dominance of approximately 90% in clinical trials conducted both across the United States and Canada, and exclusively within the United States. 

Globally, the representation of White participants in clinical trials is higher than all other ethnic groups, primarily due to the majority of trials being conducted in Western countries [12]; however, multiple factors hinder non-White participant recruitment to trials. For example, in some communities, these include poor health literacy (and by extension, health research) [89,90,91] and mistrust of scientists due to historically unethical practices, such as the Tuskegee experiment [92,93]. Additionally, within Western countries, socioeconomic deprivation is more common in non-White groups. Evidence suggests that socioeconomic factors reduce the likelihood of research participation due to work shift patterns, reduced transportation to trial sites, and reduced access to healthcare subsequently extending to trial referral [16,89,90,91,92,94,95]. The COVID-19 pandemic has also increased the awareness of health disparities in non-White participants. For example, these groups had a higher mortality rate in the United Kingdom, with evidence suggesting that this was in part due to multiple socioeconomic factors [96]. Furthermore, the lack of non-White participant representation in clinical research was also highlighted, due to devices to measure oxygen saturation levels (oximeters) not being tested adequately on non-White participants, which led to overestimation of oxygen saturation levels in non-White patients, potentially increasing fatalities in these groups [97].

Recently, a number of initiatives have been implemented to improve the representation of non-White participants in clinical trials and longitudinal research. For example, the ongoing ‘Alzheimer’s Disease Neuroimaging Initiative’ in the United States aims to recruit 60% non-White participants in its next cohort, by educating non-White groups on the importance of research [18], and the LatAM-FINGERS trial is currently conducting the first large-scale study investigating cognitive decline in Latin America, by building trust in local communities [17]. Despite these recent initiatives, such efforts and more should apply to all research designs globally. However, testing treatments across all populations at each clinical trial stage in drug development is very challenging, and would require major infrastructure and funding regulations, due to the majority of research being based in Western countries [90,91,98]. More collaborative efforts similar to this could be made with researchers in other countries, i.e., drugs that fail at earlier stages in White cohorts could be tested in groups of participants in Asia, Latin America, and Africa, in order to test their possible differential efficacies.

In trials conducted exclusively in the United States, we found that 90.6% of the participants were White, 4.9% were Black, and 4.5% were from other ethnic minority groups, despite Black and other ethnic groups, respectively, making up 13.6% and 10.6% of the population [99]. Assuming fibromyalgia is at least as prevalent in non-White populations, the proportion of ethnic minorities recruited to trials should be more than double, even when accounting for small changes to this proportion over the past 20 years [100]. Two similar analyses of trials for multiple types of cancers conducted in the United States also reported these types of discrepancies, although not as pronounced, with White and Black participant proportions reported between 82–84% and 7–11%, respectively [15,16]. To what extent these recruitment disparities can be attributed to ethnicity is complex, since socioeconomic factors affecting trial recruitment apply more to non-White populations. Of note, reduced healthcare access extending to referral to trials via a clinician is an additional socioeconomic factor in the United States, due to a private healthcare system [16,89,94,95]. 

Why are the proportions of non-White participants slightly lower in fibromyalgia trials compared with those investigating treatments for multiple types of cancer in the United States? Firstly, real prevalence differences in fibromyalgia and/or some cancers in various ethnic groups must be considered [7]. Under the diagnosis of fibromyalgia in non-White patients, extending to clinical trial recruitment is also possible. Furthermore, conflicting evidence suggests fibromyalgia prevalence is higher in both White and non-White populations [27,101]. However, the former has been attributed to both healthcare disparities and some evidence of racial discrimination when presenting with fibromyalgia symptoms to clinicians as non-White participants [27,28,29]; for a review, see Marr et al., 2020 [28]. Study type might also account for slightly lower numbers of non-White participants in fibromyalgia trials. Notably, fibromyalgia trials repurpose treatments, as opposed to cancer studies, which primarily test new drugs [15,16]; much of the FDA guidance regarding ethnic diversity is stipulated for trials seeking approval for new treatments [1,20]. 

Our longitudinal analysis found no changes to the proportion of non-White participants in multi-centre fibromyalgia trials since the start of 2000. Previous analyses of chronological changes to non-White participant proportions are limited and report mixed findings. For example, longitudinal data previously pooled from two separate analyses showed a decrease in the proportion of non-White participants in cancer trials between 1996 and 2016 [102,103]. However, a review of more recent data reported that the overall proportion of non-White participants was higher in trials published in ‘Clinical Pharmacology and Therapeutics’ between 2020 and 2021, compared with between 2000 and 2001 [2]. With initiatives to improve ethnic diversity in clinical research increasing within the past decade [1,17,18,19,20], assessing longitudinal changes to the proportion of non-White participants in trials for different conditions in more recent years could be considered important for assessing whether these initiatives are changing recruitment demographics. 

Finally, we found no significant differences in retention rates between White and non-White participants between fibromyalgia trial stages. This indicates that, at least in the context of fibromyalgia, there are no major issues with non-White participant retention in clinical research, and that previous experience of clinical trials is not a factor that affects recruitment of non-White participants to trials. To our knowledge, little research exists regarding the retention of different ethnic groups in trials investigating treatments for other conditions. However, a previous analysis investigating retention in a trial investigating antidepressant medication found that the retention of Black participants was significantly lower at some trial stages [104]. Taken together, more research is needed on ethnic minority retention in a larger range of clinical research, which also investigates retention at various trial stages. 

### Strengths and Limitations 

Several strengths and limitations of our review are noted. Firstly, our findings are not inclusive of every eligible study within our specified timeline due to journal access issues (eight publications), some trials not reporting ethnicity demographics in their metadata, and database searches potentially not picking up every published fibromyalgia trial. Nonetheless, 93% of the participants recruited across all 56 studies were from 29 multi-centre trials [52,53,54,55,56,57,58,59,60,61,62,63,64,65,66,67,68,69,70,71,72,73,75,76,77,78,79,80,81], with all but two [73,75], reporting at least the proportion of White participants. It is also unlikely that our searches missed any large RCT studies, since keywords specifying RCTs were included in searches across four databases. Taken together, it is unlikely that the missing data, primarily from smaller single-centre studies, would change our main finding of a 90% White participant dominance in fibromyalgia trials.

Secondly, in fibromyalgia research, this review identified no changes to the proportion of non-White participants since 2000, and similar trial retention rates between White and non-White ethnic groups. However, small sample sizes of 27 and four studies for our longitudinal and retention analysis, respectively, are noted; thus, we cannot strongly speculate either trend applies to non-White participants across all clinical research. 

The number of fibromyalgia trials towards the latter 2010s has dropped slightly; thus, all our findings are slightly dated and might not be inclusive of very recent changes to the recruitment of non-White participants in trials. However, our findings are strongly representative of attitudes towards ethnic group recruitment to fibromyalgia trials over the past 22 years. Notably, the impact of COVID-19 on both the numbers of recent clinical trials and potential changes to the recruitment of ethnic minorities to trials after disparities being highlighted, are areas for potential research in the coming years.

Finally, we note that all our findings are limited to pharmacological fibromyalgia studies in the United States and Canada. Follow-up reviews will be important for assessing ethnic diversity in trials conducted in other Western countries in Europe and the United Kingdom. Furthermore, assessing whether recruitment patterns are similar for non-pharmaceutical interventions in future reviews will also help to inform clinicians and researchers.

## 5. Conclusions

This review adds to the culminating literature highlighting inadequate ethnic diversity in clinical research. Furthermore, our findings indicate that the proportion of non-White participants in clinical trials in the United States should be more than double the current figure. Since the fibromyalgia studies we identified investigated the repurposing of existing treatments, further reviews could investigate whether the proportions of ethnic minorities are lower in these types of trials compared with trials testing novel pharmaceuticals. 

Additionally, our review highlights that in fibromyalgia research there has been no change to the proportion of non-White participants in trials since the start of the millennium, and almost identical retention rates between White and non-White participants. However, further research is needed to examine these trends in trials testing treatments for other neurological conditions. This could help to strengthen initiatives aimed at increasing ethnic diversity in trials.

## Figures and Tables

**Figure 1 ijerph-20-07185-f001:**
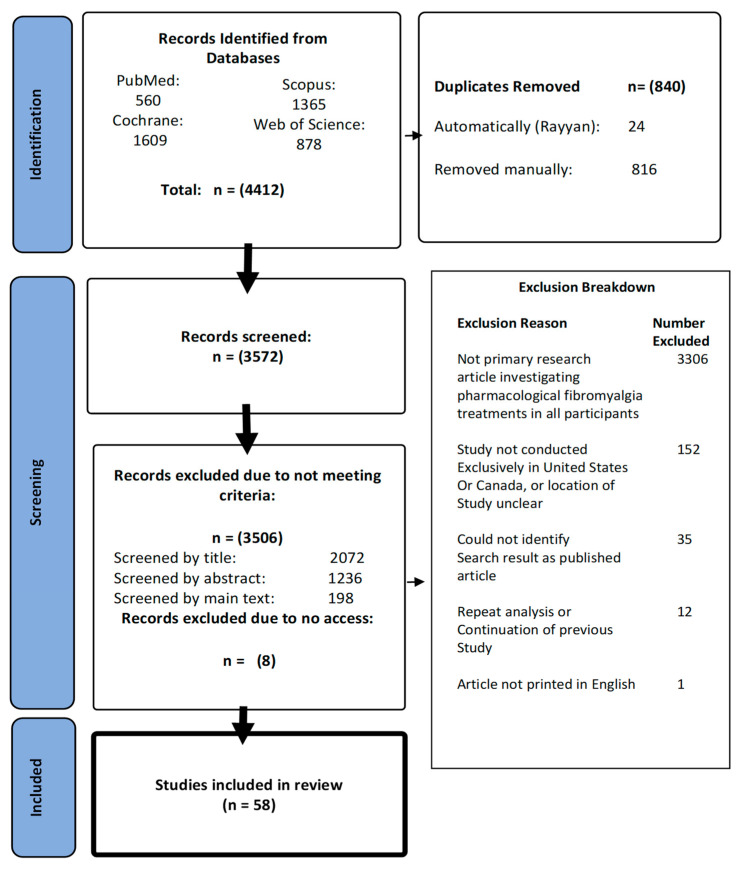
Inclusion screening process for articles identified by our searches (**left panels**), and a breakdown of exclusion reasons (**right panel**).

**Figure 2 ijerph-20-07185-f002:**
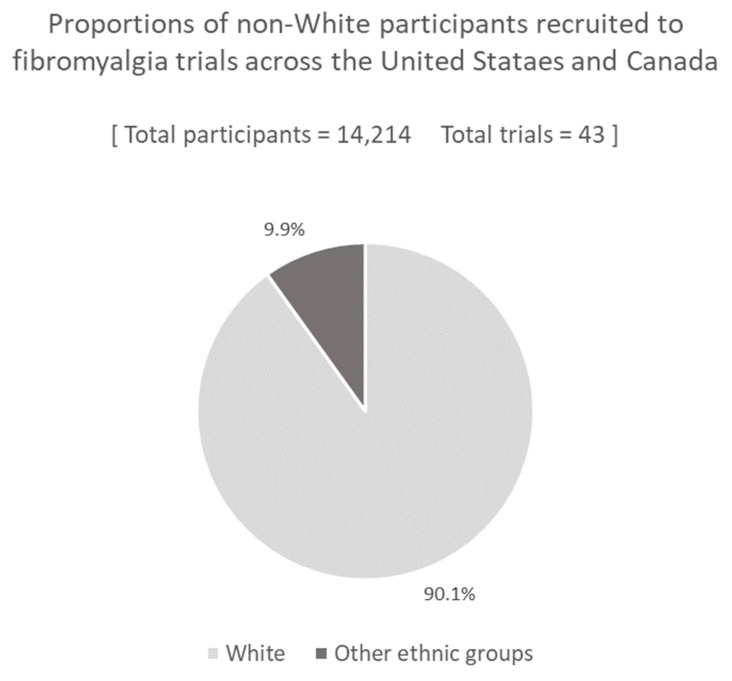
Proportion and number of White participants enrolled in fibromyalgia pharmaceutical trials compared with all other ethnic groups. Data shown are from all studies identified in our searches that recruited new participants, where the proportion of White and non-White participants was reported. Data where these classifications were not reported in these studies for individual participants accounted for <0.1% of the participants across the studies, and are not displayed on the graph.

**Figure 3 ijerph-20-07185-f003:**
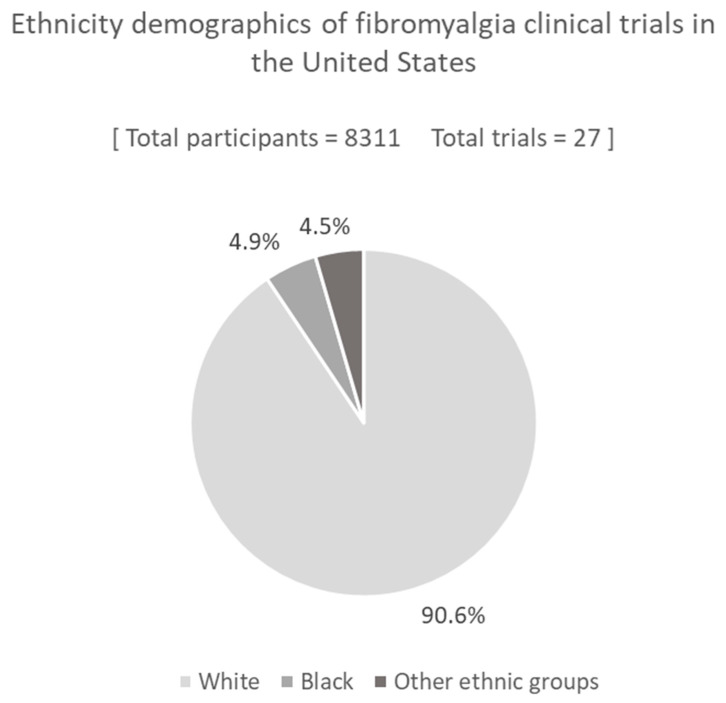
Proportion and number of White, Black, and other unspecified ethnic groups enrolled in fibromyalgia pharmaceutical trials conducted exclusively in the United States. Data from all identified United States trials where the numbers of White and Black participants were clearly specified. Data where these classifications were not reported in these studies for individual participants accounted for <0.1% of the participants across the studies, and are not displayed on the graph.

**Figure 4 ijerph-20-07185-f004:**
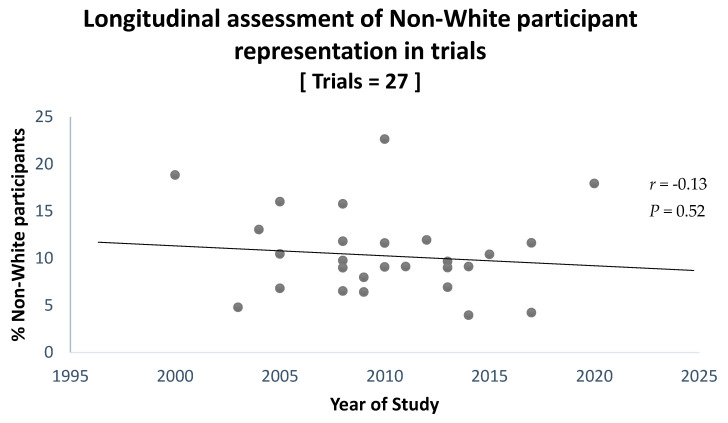
Longitudinal changes to the proportion of non-White participants in fibromyalgia trials. Pearson’s product-moment correlation coefficient found no significant changes in the proportion of non-White participants in fibromyalgia multi-centre trials between the start of 2000 to June 2022, (*r* = −0.13, [95% CI: −0.49, 0.26], *p* = 0.52, n = 27). *p*-value < 0.05 was considered significant.

**Table 1 ijerph-20-07185-t001:** Characteristics of fibromyalgia studies.

**Number of Included Articles**	
Total articles in review	58
Total trials recruiting new participants (Objectives 1–3)	56
**Participant numbers across all trials recruiting new participants**	
Total participants	14,977
Total female participants	13,591 (~90%) Data unavailable 125
Study sample size range	10–1227
Number/proportion of participants in multi-centre trials	13,996 (93.4%)
**Characteristics of trials recruiting new participants**	
Multi-centre trials (non-phase III)	26
Phase III multi-centre trials	3
Single centre trials	27

## Data Availability

Data used for analysis can be supplied from the corresponding author upon request.

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
