# Peer review of "Assessing Ethnic Minority Representation in Fibromyalgia Clinical Trials: A Systematic Review of Recruitment Demographics"

_ijerph, 2023, doi:10.3390/ijerph20247185_

Round 1

Reviewer 1 Report

Comments and Suggestions for Authors

- You identify the most relevant trials conducted in all North American countries, but you have not included trials from Mexico. What is the reason for not including trials from this North American country?

- The PRISMA flowchart does not conform to the format of the latest 2020 version. You should change the formatting to that version.

- The figures presented by the authors are of poor quality and should be changed. The information that the authors present in figure 5 is not necessary as it does not provide relevant information to the study.

Reviewer 2 Report

Comments and Suggestions for Authors

1. The authors need to rationalize their aims. For instance, why did they only include North American studies but not others (of the same/similar content) that might also provide insight into the discussion.

2. The inclusion of papers residing in the top half of Scimago ranking could significantly be misleading. Instead the authors could have preferred to include all the others and perhaps distinguished them during their discussion. Excluding articles just based on the journal IF would be dangerous.

3. Shapiro Wilk test is misspelled.

4. Conclusion section could be more concise.

Reviewer 3 Report

Comments and Suggestions for Authors

The current review manuscript assesses the ratios of non-White and White participants in fibromyalgia who participated in pharmaceutical trials. This topic is novel, as far as I know, and has an impact on our research and clinical field of fibromyalgia.

Here are some points of discussion for an informative article for readers such as clinicians and researchers in clinical trials.

It would be interesting to see any difference in terms of the findings when compared with non-pharmaceutical clinical trials, such as with cognitive-behavioral therapy. This will provide some clues if the disproportionality is also observed in different clinical trial fields.

It could’ve been interesting if there existed a known natural prevalence of fibromyalgia as a reference. This will help to discuss if the recruitment in the trials well represents the prevalence of the disorder and if there are other factors that might affect the difference, if any.

It would be informative to discuss more on the changes in ratio by year. Since 2000 there have been several national/global issues, such as COVID-19 pandemic, and as the author indicated in the discussion (“the number of fibromyalgia trials within the last five years had dropped”), the relationship between the total number of trials and participants by year and possible national (U.S.) or global issues might suggest some insights.

A minor point is about the missing data in Figures 2 and 3. Shouldn’t it be screened beforehand? I was not sure what the missing data meant after the screening and why it happened.

Round 2

Reviewer 2 Report

Comments and Suggestions for Authors

Revised well.